# Multiple Preprocessing Hybrid Level Set Model for Optic Disc Segmentation in Fundus Images

**DOI:** 10.3390/s22186899

**Published:** 2022-09-13

**Authors:** Xiaozhong Xue, Linni Wang, Weiwei Du, Yusuke Fujiwara, Yahui Peng

**Affiliations:** 1Information and Human Science, Kyoto Institute of Technology University, Kyoto 6068585, Japan; 2Retina & Neuron-Ophthalmology, Tianjin Medical University Eye Hospital, Tianjin 300084, China; 3School of Electronic and Information Engineering, Beijing Jiaotong University, Beijing 100044, China

**Keywords:** optic disc segmentation, multiple preprocessing, hybrid level set, wide-angle fundus images, four-side evaluation

## Abstract

The accurate segmentation of the optic disc (OD) in fundus images is a crucial step for the analysis of many retinal diseases. However, because of problems such as vascular occlusion, parapapillary atrophy (PPA), and low contrast, accurate OD segmentation is still a challenging task. Therefore, this paper proposes a multiple preprocessing hybrid level set model (HLSM) based on area and shape for OD segmentation. The area-based term represents the difference of average pixel values between the inside and outside of a contour, while the shape-based term measures the distance between a prior shape model and the contour. The average intersection over union (IoU) of the proposed method was 0.9275, and the average four-side evaluation (FSE) was 4.6426 on a public dataset with narrow-angle fundus images. The IoU was 0.8179 and the average FSE was 3.5946 on a wide-angle fundus image dataset compiled from a hospital. The results indicate that the proposed multiple preprocessing HLSM is effective in OD segmentation.

## 1. Introduction

The optic disc (OD) is a region where blood vessels and optic nerves pass through the retina [1]. The area ratio of OD to optic cup (OC) is the main criterion of glaucoma diagnosis [2]. The OD position is the benchmark for determining the extent of the macula [3], for example, the macula always presents on the left side of OD for the right eye, and on the right side of the OD for the left eye. Moreover, macular edema can be used to diagnose diabetic retinopathy (DR) [4]. Therefore, accurate OD segmentation may play important roles in many fundus disease diagnoses such as glaucoma [5] and DR [6].

OD is a highly bright oval-shaped yellowish region [7] in fundus images. There are two types of fundus images that are distinguished on the basis of the field of view (FOV) [8,9]. One is posterior part fundus images that have a narrow FOV, as shown in Figure 1. The FOV is from $30^{\circ }$ to $60^{\circ }$ [10,11,12,13]. The other type is wide-angle fundus images with a much wider FOV, as shown in Figure 2. The FOV is from $130^{\circ }$ to $200^{\circ }$ [14,15]. Accurate OD segmentation in fundus images is a challenging task for the following reasons (Figure 3): (1) Some parts of OD boundaries are covered by blood vessels; (2) some types of noise such as parapapillary atrophy (PPA) and bright area affect OD segmentation. For Reason (1), blood vessels can be removed by using a mathematical morphology [16]; however, some bright noise and PPA are mixed in the OD area, resulting in undersegmentation. For Reason (2), according to the different textures between PPA and other regions, a gray level co-occurrence matrix (GLCM) can be used to detect the PPA [17]. However, it is easy for misdetections or false detections to occur when the texture features of PPA are not obvious. Therefore, the effects of bright noise and PPA regions in OD segmentation are still unresolved.

This paper proposes a multiple preprocessing hybrid level set model (HLSM) to solve the problems in which OD segmentation is affected by bright noise and PPA. The contour in HLSM is mainly controlled by area-based and shape-based terms. The area-based term represents the difference of the average pixel values between the inside and outside of the contour, while the shape-based term measures the distance between a prior shape model and the contour. Moreover, this paper proposes a novel evaluation method of four-side evaluation (FSE), and addresses the limitation of objective evaluation methods such as intersection over union (IoU) in over- and undersegmentation around the entire OD. Furthermore, the FSE is a subjective evaluation from clinicians that can prove whether the segmentation results are accurate enough in the view of clinician. The effectiveness of the proposed approach is verified in two types of fundus image, posterior and wide-angle. The proposed method achieved results with an average IoU of 0.9275 and an average FSE of 4.6426 in the DRISHTI-GS posterior fundus image dataset, and an average IoU of 0.8179 and average FSE of 3.5946 in the TMUEH wide-angle fundus image dataset.

The contributions of this paper are as follows:1.The proposed method has higher robustness that could successfully segment the OD in both the posterior and wide-angle fundus images.2.The effect of PPA and bright noise causes undersegmentation, and the OD being occluded by blood vessels causes undersegmentation. The proposed approach achieves high accuracy in OD segmentation results by solving these problems.3.A new evaluation method, FSE, is proposed for clinicians to subjectively evaluate OD segmentation results.

This paper is composed as follows: Section 2 briefly introduces some research background on OD segmentation in fundus images; Section 3 proposes and explains the multiple preprocessing hybrid level set model; Section 4 outlines the experiment and discussion; Section 5 draws some conclusions.

## 2. Related Work

Proposed OD segmentation algorithms can be roughly divided into five categories: threshold- [18,19], pattern- [20,21,22,23], classification and clustering- [24,25,26], active contour model- [16,27,28,29,30,31,32,33], and deep learning- [34,35,36]-based methods. In threshold-based methods, the ODwas segmented by OTSU and other thresholding methods in [18,19]. Although they adopted multichannel information and achieved fast processing speed, it was susceptible to bright noise around the OD, and the segmentation accuracy was low. In pattern-based methods, a limited ellipse fitting method was used to segment the OD [20]. However, the OD is not completely elliptical, so this method was not accurate enough. Segmenting the OD with an active shape model is easily effected by the PPA and bright noise [21,22,23]. Regarding classification and clustering-based methods, in [24], superpixels were first segmented according to the similarity between each pixel and then classified with a random forest method. The K-means clustering method [25] and density-based spatial clustering method [26] are also used to segment the OD. However, both classification- and clustering-based methods require pre- and postprocessing, and cannot dispose of the effects of PPA and blood vessels. With the rapid development of deep learning in recent years, it has also been applied to OD segmentation. In deep learning-based methods, U-Net combines encoding and decoding information; due to this characteristic, U-Net, improved U-Net, or structures inspired by U-Net are widely used in OD segmentation. For example, U-Net was directly trained to segment the OD in [37,38]. A deep-learning structure, M-Net, was developed in [34] that combines features in polar coordinates and Euclidean coordinates. U-Net and ResNet block were combined in [36] and [39], and U-shaped DenseNet was introduced in [40]. NENet, which was inspired by U-Net, was utilized to segment OD in [41]. In addition to the U-shaped network, a segmentation adversarial network (SAN) based on generative adversarial networks (GANs) was proposed in [35], and a fully connected network was improved by combining the distance and density features in [42]. A structure that combines DeepLabv3+ and MobileNet was proposed to segment the OD in [43]. However, it is difficult to obtain a large amount of training data, which is the difficult aspect of deep-learning methods in OD segmentation.

Because of the high robustness and fewer parameters, the active contour model is also widely applied in OD segmentation. Some researchers proposed various types of active contour models, for example, variation-based [44], gradient-based [45], area-based [46] and shape-based [47]. In active-contour-model-based methods, the gradient-based level set model was used in [27,29,30,31]. However, segmentation fails if the OD boundary is extremely smooth. The level set model was improved in [16,28] by applying multiple energy functionals that included gradient-, area-, and shape-based energy functionals. A shape-based functional is able to limit the shape of the contour, the area-based functional is adopted to change the position and size of the prior shape model, and the gradient-based functional is the main source of energy to find the OD boundaries. This method solves the above problem of OD segmentation failure, but the changed shape model is easily effected by PPA and bright noise. The area-based level set model was used in [32,33], which mainly drove the contour according to the average pixel values inside and outside the contour. However, the contour is sensitive to blood vessels, PPA, and bright noise if only using an area-based energy functional, resulting in a reduction in segmentation accuracy.

## 3. Multiple Preprocessing Hybrid Level Set Model

Because of the problems of OD segmentation being affected by PPA, bright noise, and blood vessels, this paper proposes a multiple preprocessing HLSM. A morphological method is able to avoid OD segmentation from being affected by blood vessels; the hybrid level set method with shape constraints is used to control the evolution of the contour and avoid the effect of bright noise; adaptive threshold-based ellipse fitting can find a more appropriate initial value of level set function and the effect of PPA can be simultaneously avoided. Figure 4 shows the flowchart of the proposed approach. In this section, “contour” means the contour of zero-level set, and “boundary” means the boundary of OD.

### 3.1. Hybrid Level Set Model

The level set model is robust and flexible. It is not sensitive to small noise and can flexibly change the model to be suitable for specific tasks. So, we designed a model that is based on the level set model to segment OD. The level set model is a type of active contour model [45] that embeds the contour into a level set function ϕ that has a higher dimensional number and uses a certain level set (usually zero-level set) to represent the current contour. The activity of the contour is shown in Equation (Equation 1). *F* is the energy function, and for different tasks, the corresponding *F* needs to be designed. $\left|\nabla \phi \right|$ is the gradient of level set function. This equation means that level set function ϕ changes along the gradient direction under the action of energy function *F*. The change in unit time *t* is the product of *F* and $\left|\nabla \phi \right|$.
(1)$$\frac{\partial \phi }{\partial t}=F\left|\nabla \phi \right|$$

For solving the problems in OD segmentation, the corresponding *F* should have the following characteristics: (1) Because some areas in OD are occluded by blood vessels, OD segmentation results have a depression in these areas. To avoid the effect of blood vessels, the smoothness of the contour should be controllable. (2) There are some PPA regions and bright noise surrounding OD. To avoid being affected by this noise, the contour evolution should be limited. (3) Due to the existence of PPA regions and bright noise, the OD contrast is low, and the gradient of the boundary is poor. So, segmentation failure caused by a poor gradient should be avoided.

According to the above characteristics, as shown in Equation (Equation 2), this paper proposes a level set model that has four terms: the distance-regularized, line integral and area integral, area-based, and shape-based terms, which is why it is called HLSM. The most important terms in HLSM are the area- and shape-based terms. The area-based term is used to avoid segmentation failure in low-contrast images. The shape-based term is utilized to avoid the effects of bright noise and PPA regions. Apart from this, the distance-regularized term is used to avoid reinitialization, and the line and area integral terms are able to control the contour smoothness and evolutionary direction, respectively.
(2)$$\begin{array}{cc} F & =F_{distance-regularized}\\ \; & +F_{line\;integral\;\&\;area\;integral}\\ \; & +F_{area\_based}\\ \; & +F_{shape\_based} \end{array}$$

The distance-regularized term is represented by Equation (Equation 3), where μ is the weight of this term, Ω is the image domain, (x,y) is each point in the image. Since the signed distance function (SDF) has the property of $\left|\nabla \phi \right|=1$, the contour is usually embedded in SDF for level set model. However, when it is applied to an image, SDF needs to be discretized, so that the level set function no longer remains as the SDF in the process of evolution. The level set function needs to be reinitialized after each evolution. The distance-regularized term can keep $\left|\nabla \phi \right|$ near 1, and keep the level set function as an SDF as much as possible. Thus, this term is able to avoid reinitialization and speed up the evolution.
(3)$$F_{distance-regularized}=\mu \int_{\Omega }^{\;}p\left(\left|\nabla \phi \right|\right)dxdy$$
where,
(4)$$p\left(s\right)=\left\{\begin{array}{c} \frac{1}{2\pi }sin\left(2\pi s\right),\;if\;s\leq 1\\ s-1,\;if\;s>1 \end{array}\right.$$

The line integral term is expressed in Equation (Equation 5), while the area integral term is expressed in Equation (Equation 6). α is the weight of the line integral term, β is the weight of the area integral term, and function δ is the differential of function *H*. The line integral term is able to control the smoothness of the contour. As shown in Figure 5b, the line integral term was applied, while in Figure 5a, α was set to 0. The blue contour in Figure 5b was evidently smoother than the blue contour in Figure 5a. The area integral term controlled the evolutionary direction.
(5)$$F_{line\;integral}=\alpha \int_{\Omega }^{\;}\delta \left(\phi \right)\left|\nabla \phi \right|dxdy$$
(6)$$F_{area\;integral}=\beta \int_{\Omega }^{\;}H(\phi )dxdy$$
where,
(7)$$H_{\varepsilon }\left(x\right)=\left\{\begin{array}{c} 1,\;if\;x>\varepsilon \\ 0,\;if\;x<-\varepsilon \\ \frac{1}{2}\left[1+\frac{x}{\varepsilon }+\frac{1}{\pi }sin\left(\frac{\pi x}{\varepsilon }\right)\right],if\;\left|x\right|\leq \varepsilon \end{array}\right.$$
(8)$$\delta_{\varepsilon }\left(x\right)=H_{\varepsilon }^{{}^{\prime }}\left(x\right)=\left\{\begin{array}{c} 0,if\;|x|>\varepsilon \\ \frac{1}{2\varepsilon }\left[1+cos\left(\frac{\pi x}{\varepsilon }\right)\right],if\;\left|x\right|\leq \varepsilon \end{array}\right.$$

The formula of the area-based term is shown in Equation (Equation 9), where $u_{0}$ is the average pixel values of the whole image, and $c_{1},c_{2}$ are the average pixel values inside and outside the contour, respectively. The area-based term evolves the contour through the difference of the average pixel values between the inside and outside of the contour, and the weight of inside and outside can be controlled by $\lambda_{in}$ and $\lambda_{out}$. As shown in Figure 6, when the boundary of an object is fuzzy, the gradient of the boundary is low. If using the gradient-based energy functional, the contour does not converge or even disappear. The area-based term is based on the pixel difference; therefore, it can achieve excellent performance in low-contrast and low-gradient images.
(9)$$\begin{array}{cc} F_{area\_based} & =\lambda_{in}\int_{\Omega }^{\;}{\left|u_{0}-c_{1}\right|}^{2}H\left(\phi \right)dxdy\\ \; & +\lambda_{out}\int_{\Omega }^{\;}{\left|u_{0}-c_{2}\right|}^{2}\left(1-H(\phi )\right)dxdy \end{array}$$

The shape-based term is expressed in Equation (Equation 10), where $\lambda_{shape}$ is the weight of this term, and $\phi_{initial}$ is the initial value of level set function. The shape-based term measures the distance between the current contour and shape model, thereby limiting the contour evolution. In this paper, the shape model was the initial value of level set function. Thus, if a great initial value is detected, the HLSM can avoid the effects of PPA regions and bright noise under the action of the shape-based term. Likewise, if the detected initial value contains PPA and bright noise, these regions cannot be ignored by HLSM, resulting in oversegmentation. This means that the initial value is extremely important for HLSM. The detection of the initial value is introduced in Section 3.2.4. As shown in Figure 7a, $\lambda_{shape}$ was set as 0, and in Figure 7b, the shape-based term was utilized; the segmentation result in Figure 7a was affected by bright noise.
(10)$$F_{shape\_based}=\lambda_{shape}\int_{\Omega }^{\;}{\left(H\left(\phi \right)-H\left(\phi_{initial}\right)\right)}^{2}dxdy$$

Lastly, the energy functional used in this paper is shown in Equation (Equation 11).
(11)$$\begin{array}{cc} F & =\mu \int_{\Omega }^{\;}p\left(\left|\nabla \phi \right|\right)dxdy\\ \; & +\alpha \int_{\Omega }^{\;}\delta \left(\phi \right)\left|\nabla \phi \right|dxdy+\beta \int_{\Omega }^{\;}H(\phi )dxdy\\ \; & +\lambda_{in}\int_{\Omega }^{\;}{\left|u_{0}-c_{1}\right|}^{2}H\left(\phi \right)dxdy\\ \; & +\lambda_{out}\int_{\Omega }^{\;}{\left|u_{0}-c_{2}\right|}^{2}\left(1-H(\phi )\right)dxdy\\ \; & +\lambda_{shape}\int_{\Omega }^{\;}{\left(H\left(\phi \right)-H\left(\phi_{initial}\right)\right)}^{2}dxdy \end{array}$$

### 3.2. Multiple Preprocessing

Although the level set model is highly robust, it is also easier to obtain highly accurate segmentation results from an image with less noise and higher contrast. Furthermore, the proposed HLSM requires an initial value that can exclude PPA regions and bright noise. Therefore, multiple preprocessing includes the detection of multiple feature-based regions of interest (ROI), quantitative-analysis-based channel selection, morphological-based blood vessel removal, frequency domain-based noise removal, and initial-value detection with adaptive threshold-based ellipse fitting.

#### 3.2.1. Region of Interest (ROI) Detection

OD only occupies a small part of fundus images; in order to speed up OD segmentation and reduce the effect of noise, it is necessary to locate the OD and detect the ROI. The OD is an oval-shaped area with high vascular density and high brightness in fundus images, and these characteristics can be used to accurately locate OD [48]. The side length of the extracted ROI is 3 times the average diameter of OD. Because there is no guarantee that the center of OD would be detected, and there are differences in the size of the OD, it is necessary to leave free space around the OD. Therefore, the size of the ROI is 3 times the average diameter of the OD. The flowchart of ROI detection is shown in Figure 8, the results of each process in posterior fundus images are shown in Figure 9, and the results of each process in wide-angle fundus images are presented in Figure 10. In wide-angle fundus images, a circular mask image (Figure Figure 10a) is used to avoid the influence of eyelashes and the fundus camera wall. Each process of ROI detection is explained in detail as follows.

Step 1: Extracting the green channel (Figure 9b and Figure 10b) from the RGB color space and inverting it (Figure 9c and Figure 10c). In order to take advantage of the feature of high vascular density, it is necessary to roughly segment the blood vessels. In fundus images, blood vessels have a high contrast in the green channel, which is why the green channel is extracted. However, blood vessels are darker than those in other areas in the green channel. Thus, for using morphological top-hat transformation to segment blood vessels, the green channel is inverted.Step 2: Using morphological top-hat transformation to segment blood vessels (Figure 9d and Figure 10d). Although the obtained blood-vessel map is not clear and accurate, it is fast and adequate enough to find the areas with high vascular density.Step 3: Finding several circular areas with the highest vascular density (Figure 9e and Figure 10e). The radius of circular areas is the average OD radius. Because of the rough and imprecise blood-vessel map, selecting only a few areas may not include OD. Thus, 20 circular areas were selected in this paper.Step 4: Selecting one area (Figure 9f and Figure 10f) which has the largest number of 2% brightest pixels in value channel. The purpose of this step is to utilize the characteristic of high brightness.Step 5: Extracting the rough ROI (Figure 9g and Figure 10g) by this circular area. The side length of rough ROI is 4 times the average diameter of OD and the center of rough ROI is the center of circular area. As shown in Figure 11, there was still an error (the OD was not located at the center of ROI image) if locating OD only on the basis of high vascular density and high brightness.Step 6: Correcting the ROI (Figure 9h and Figure 10h) by Hough Circular Transform from rough ROI. This step uses the characteristic of oval-shaped. The side length of corrected ROI is 3 times the average diameter of OD.

**Figure 8 sensors-22-06899-f008:**
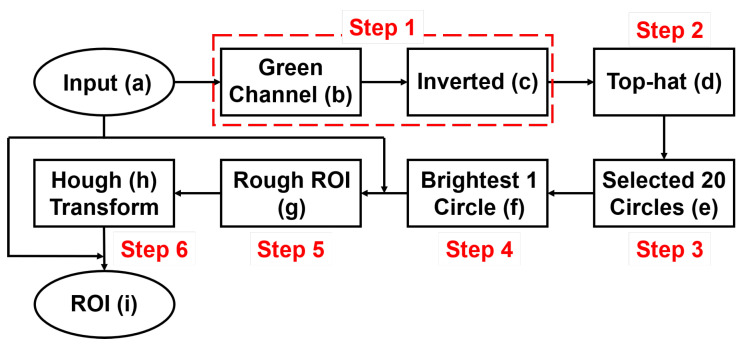
Flowchart of ROI detection: (**a**–**i**) correspond to images in Figure 9 and Figure 10.

**Figure 9 sensors-22-06899-f009:**
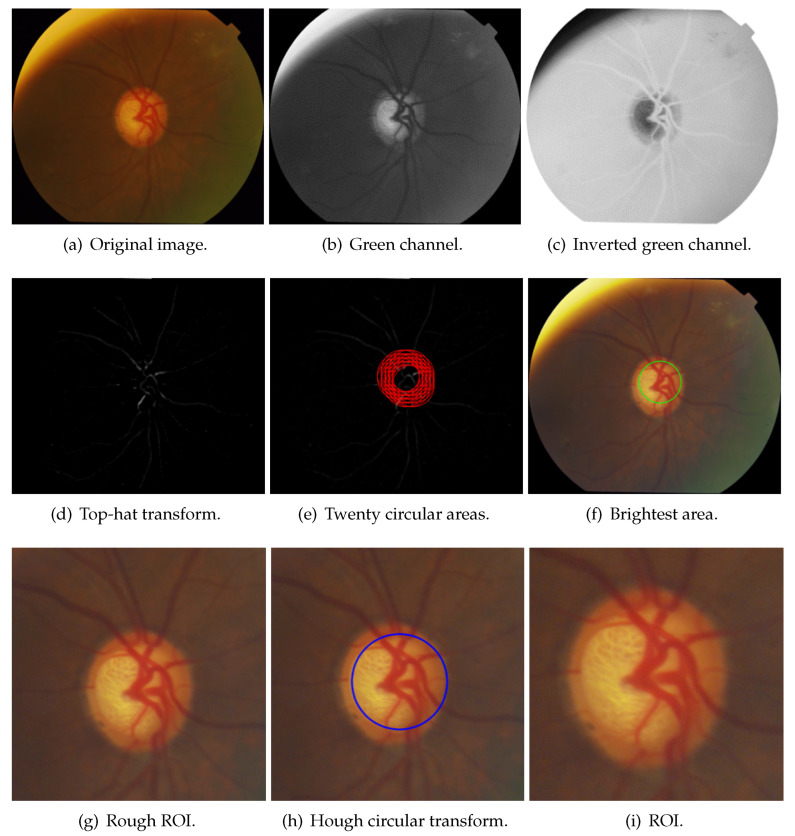
Results of ROI detection in posterior fundus images.

**Figure 10 sensors-22-06899-f010:**
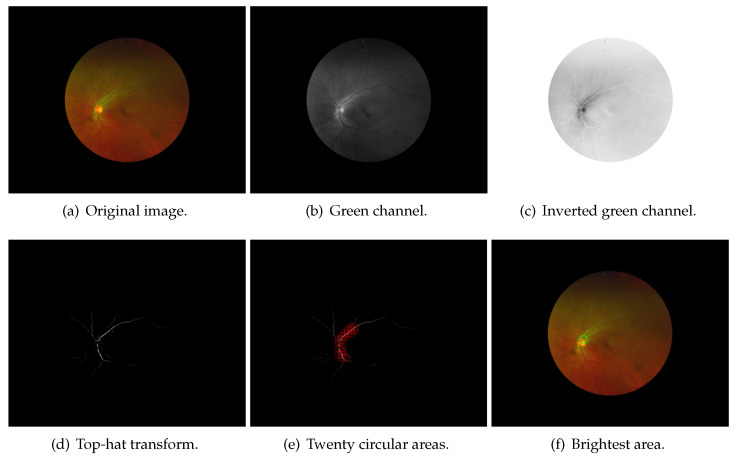
Results of each process of ROI detection in wide-angle fundus images.

**Figure 11 sensors-22-06899-f011:**
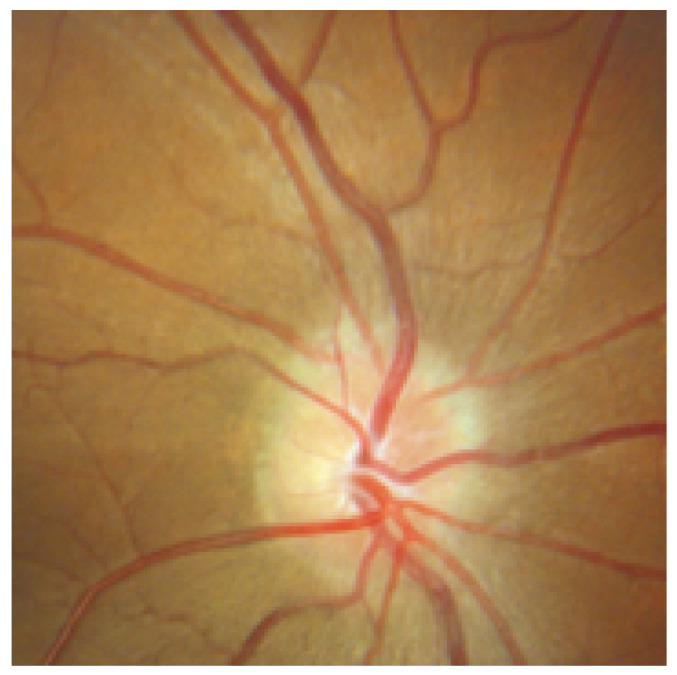
Rough ROI with poor result.

#### 3.2.2. Channel Selection

Tissues appear with different intensities in different color space; thus, selecting an appropriate color channel with a higher contrast is useful in obtaining more accurate OD segmentation results. The authors in [49,50] proposed that the red channel in RGB color space is the most suitable for OD segmentation but without justification. This paper quantitatively analyzes the channels in RGB, HSV, and LAB color spaces through the contrast-to-noise ratio (CNR, value in Equation (Equation 12)).
(12)$$\mathrm{CNR}=\frac{\left|{\mathrm{mean}}_{\mathrm{foreground}}-{\mathrm{mean}}_{\mathrm{background}}\right|}{{\mathrm{std}}_{\mathrm{background}}}$$
where ${\mathrm{mean}}_{\mathrm{foreground}}$ is the average pixel values of the foreground, and ${\mathrm{mean}}_{\mathrm{background}}$ and ${\mathrm{std}}_{\mathrm{background}}$ are the mean and standard deviation pixel values of background, respectively. In this paper, the foreground is the OD area, while the background is the areas except OD.

A large CNR value means that the difference of the average pixel values between the foreground and background is large, and the difference of the background pixel values is low. Therefore, if the CNR value of an image is large, it is a high-contrast and low-noise case.

The images from the DRISHTI-GS dataset were used to calculate the CNR, and results are shown in Table 1. The average CNR in the value channel was 5.0408, and red channel was 4.9824, which were significantly higher than those of other channels, but the variance CNR in the value channel was 5.4443, which was slightly lower than that in the red channel, 5.4550. Thus, the value channel was selected to segment OD in this paper.

#### 3.2.3. Blood-Vessel and Noise Removal

The flowchart of blood-vessel and noise removal is shown in Figure 12, while Figure 13 shows the result of each process. First, the Gaussian filter was used on the value channel(Figure 13a) to remove some small amount of noise. The result after Gaussian filter is Figure 13b. The size of the Gaussian kernel in this paper was 17×17, and the deviations in the horizontal and vertical directions were both 1. Then, blood vessels were removed with morphological closing and opening (Figure 13c). The used structure element in the morphological process is an ellipse with the size of 15×15, and the iteration times were 3. Here, the product of size and iteration times should be larger than the thickness of the blood vessels. As shown in Figure 14, some high-frequency noise is introduced after morphological processing. Lastly, a low-pass filter was used to remove this high-frequency noise (Figure 13d). The mask used in the low-pass filter was a 40×40 square in the center.

However, as shown in Figure 15, the process of blood-vessel and noise removal generates some bright noise that connects with the OD area, resulting in a low contrast of the OD boundary and undersegmentation. This problem is solved with a more suitable initial value of the level set function. This bright noise is ignored under the constraint of shape-based term in the HLSM.

#### 3.2.4. Initial-Value Detection

According to the shape-based energy functional (Equation (Equation 10)), the initial value significantly influences the segmentation results in HLSM. Furthermore, a suitable initial value can reduce the number of iterations and speed up the convergence of the contour. In this paper, the adaptive threshold-based ellipse fitting is proposed to obtain the initial value. Figure 16 shows the flowchart of initial value detection. Figure 17 shows the result of each process.

The value channel (Figure 17a) in the HSV color space is processed to obtain the initial value. First, the binary image (Figure 17b) is received by using the adaptive threshold (Equation (Equation 13)). I is the original image, and $I_{\mathrm{binary}}$ is the obtained binary image. w,h are the width and height of the batch, respectively. In this paper, wandh are the half width and height of ROI, respectively. (x,y) is each point in the image, while $\left(x^{{}^{\prime }},y^{{}^{\prime }}\right)$ is each point in the batch. *C* is the offset that was set as 0 in this paper. Then, the small links between each connected area are disconnected with morphological opening (Figure 17c). Next, the maximal 4-connected area is selected (Figure 17d), and its boundary is detected (Figure 17e). Lastly, the initial value (Figure 17f) is generated with ellipse fitting from this boundary. OD is oval-shaped in fundus images, and an ellipse-shaped initial value can render the segmentation result roughly elliptical; therefore, the segmentation result is not affected by bright noise. As shown in Figure 18, there are some PPA regions in Figure 18a, the Figure 18b and Figure 18c are the binary image after adaptive thresholding and the image of maximal 4-connected area, respectively. These PPA regions are removed in initial-value detection, as the detected initial value excludes PPA regions. Under the action of a shape-based term, segmentation results are not affected by PPA regions.
(13)$$I_{\mathrm{binary}}\left(x,y\right)=\left\{\begin{array}{c} 1,\mathrm{if}\frac{\sum_{x^{{}^{\prime }}=x-\left\lfloor \frac{w}{2}\right\rfloor }^{x+\lfloor \frac{w}{2}\rfloor }\sum_{y^{{}^{\prime }}=y-\left\lfloor \frac{h}{2}\right\rfloor }^{y+\lfloor \frac{h}{2}\rfloor }I\left(x^{{}^{\prime }},y^{{}^{\prime }}\right)}{w\times h}-C\leq I\left(x,y\right)\\ 0,\mathrm{else} \end{array}\right.$$

## 4. Experiment

The effectiveness of the proposed multiple preprocessing HLSM was verified in two datasets: DRSHTI-GS and Tianjin Medical University Eye Hospital (TMUEH). Furthermore, the segmentation results were evaluated with two different evaluation methods, namely, intersection over union (IoU) and four-side evaluation (FSE), and were compared with other approaches.

### 4.1. Data Sets

There are two types of widely used fundus images: posterior and wide-angle fundus images. A posterior fundus image can clearly show the fundus because it is generated by a white light source [51]. However, it can only display the fundus in a narrow FOV [52]. Wide-angle fundus images can demonstrate the fundus in a wide FOV, but their quality is lower than that of posterior fundus images because wide-angle fundus images are only generated by red and green light sources. Since posterior fundus images have a history of more than 150 years [53], there are many public posterior-fundus-image datasets, such as MESSIDOR [54], ORIGA-light [11], DRIONS-DB [55], and DRISHTI-GS [13]. Because of the high resolution and reliable ground truth (GT), DRISHTI-GS was applied in this paper.

**DRISHTI-GS** [13]: a widely used public dataset in OD segmentation that contains 101 posterior fundus images (Figure 19). Each fundus image is centered on the OD, and the FOV is $30^{\circ }$. The resolution ranges from 2047×1745 to 2468×1762. The GT for OD and OC boundaries was marked by four clinicians with 3, 5, 9, and 20 years of clinical experience.

**Tianjin Medical University Eye Hospital (TMUEH)**: currently, there is no public wide-angle fundus image dataset, so some fundus images were required from TMUEH. It is a non-public dataset. It contains 37 wide-angle fundus images (Figure 20). Each fundus image is centered on the fovea, and the FOV is $200^{\circ }$. The resolution is 3900×3072. The OD GT was marked by a clinician with 17 years of clinical experience.

### 4.2. Evaluation Criteria

This paper both subjectively and objectively evaluates the segmentation results. The subjective evaluation was FSE and was conducted by clinicians to effectively evaluate whether the segmentation results were clinically meaningful. The value of IoU was used for the objective evaluation, which is expressed by Equation (Equation 14). ${\mathrm{area}}_{\mathrm{result}}$ and ${\mathrm{area}}_{GT}$ are the area of segmentation result and the GT, respectively.
(14)$$\mathrm{IoU}=\frac{{\mathrm{area}}_{\mathrm{result}}\cap {\mathrm{area}}_{GT}}{{\mathrm{area}}_{\mathrm{result}}\cup {\mathrm{area}}_{GT}}$$

IoU represents the overlapping ratio between the segmentation result and GT. A higher IoU value means a better segmentation result. However, there is a problem in objective evaluation methods such as IoU. When the surroundings of the OD are under- or oversegmented, as shown in Figure 21, the Figure 21a is an example of overall oversegmentation, the Figure 21b is an example of partial oversegmentation, the IoU value is higher in Figure 21a, but the segmentation result is too large. This can be regarded as OD boundaries not being found at all. Furthermore, this result greatly impacts subsequent diagnosis, such as glaucoma misdiagnosis through the area ratio of OD to OC. Due to the above problem in IoU, the following subjective evaluation method was designed.

First, subjective evaluation criteria were established; then, the clinicians scored each OD segmentation result according to the criteria. As shown in Figure 22, the OD is divided into four parts: superior, nasal, inferior, and temporal. Each part is $90^{\circ }$. Depending on the opinion of clinicians, one point is given if each part was segmented accurately enough. The specific evaluation criteria are shown in Table 2. Since the evaluation method divided OD into four parts for evaluation, it is called the FSE.

The FSE proposed in this paper evaluates the OD segmentation results from another perspective. For example, for the two segmentation results of the same image in Figure 21, In Figure 21a, better scores could be obtained regardless of IoU, dice value [36], or mean square error [56], while only 0 or 1 could be obtained in FSE. In Figure 21b, the score was lower when using IoU, but higher scores could be obtained when using FSE. In other words, objective evaluation methods such as IoU focus more on the whole OD, while the FSE proposed in this paper focuses more on each part of the OD.

### 4.3. Parameters

The parameters of the proposed HLSM are shown in Table 3. The main parameters are the weights of each energy functional term. The average radius of OD is used in ROI detection, which is used to determine the size of an ROI image.

### 4.4. Experimental Results

#### 4.4.1. Segmentation Results

The segmentation results of posterior and wide-angle fundus image are shown in Figure 23 and Figure 24, respectively. The evaluation results with IoU are shown in Table 4, and the results evaluated by FSE are displayed in Table 5. The maximal, minimal, average, variance, and distribution of IoU are illustrated in Table 4, and the average and distribution of FSE are illustrated in Table 5.

As shown in Table 6, the results of IoU evaluation in the DRISHTI-GS dataset were compared with those in other algorithms. In [57,58,59], the U-Net architecture was used to segment OD. The authors in [60] used a boundary and entropy-driven adversarial learning-based deeplabv3+ (BEAL-Deeplabv4+) architecture to segment the OD. In [61], entropy information was also used; the architecture was an entropy sampling- and ensemble learning-based CNN (EE-CNN). In [62], many common non-deep-learning methods were reproduced to segment the OD. A method called level set adaptively regularized kernel-based intuitive fuzzy C means (LARKIFCM) was also proposed to segment OD.

In order to compare the OD segmentation results in the TMUEH dataset and the results with FSE evaluation method, two algorithms from other papers were reproduced: one using the active contour-based method [31], and the other using the threshold-based method [18]. The proposed ROI detection method was used when reproducing these two methods. The segmentation results evaluated with IoU are displayed in Table 7 and Table 8, and the results evaluated with FSE are shown in Table 9 and Table 10.

#### 4.4.2. Discussion

In the DRISHTI-GS and TMUEH datasets, the average IoU achieved 0.9257 and 0.8179, and the average FSE achieved 4.6436 and 3.5946, respectively. In the DRISHTI-GS dataset, 99 in 101 cases (98%) had an IoU value of more than 0.8, and 83 in 101 cases (82%) had 5 points with the FSE evaluation method. In the TMUEH dataset, 28 in 37 cases (76%) had an IoU value of more than 0.8, and 29 in 37 cases (78%) had 3 or more points with the FSE evaluation method.

As shown in Figure 23 and Figure 24, there was no depression or there was only a little depression in the blood-vessel area. This proves that the morphological-processing-based blood-vessel removal and frequency-domain-based noise removal were effective. In Figure 23b and Figure 24b, there was bright noise surrounding the OD in these cases due to the constraint of the shape-based term on the contour evolution. The HLSM avoided these effects and achieved outstanding segmentation results. In Figure 23c and Figure 24c, the PPA regions were also ignored in these cases. This demonstrates that the adaptive-thresholding-based ellipse fitting was able to detect a more suitable initial value. Furthermore, under the constraint of the shape-based term, the effect of the PPA regions was avoided. These segmentation results prove that the problems of occlusion by blood vessels, and the effects by bright noise and PPA regions were solved with the proposed multiple preprocessing HLSM.

As shown in Table 6, the proposed multiple preprocessing HLSM achieved better accuracy than that of other recent algorithms. As shown in Table 7 and Table 8, the proposed model achieved a higher average IoU and better distribution than those of the two reproduced algorithms in the TMUEH dataset. Among these proposed algorithms, the segmentation results were affected by blood vessels in [58,59,61,62], while the proposed approach avoided this effect with a morphological process. A postprocessing convex hull was utilized in [61]; thus, it was also affected by dark noise. The effect of PPA also existed in [59]. The segmentation results in [60] were mainly affected by bright noise. However, the proposed approach was able to avoid the effect of PPA and bright noise with a suitable initial value and the constraint of the shape model. Due to the low contrast and effect of bright noise, both [31] and [18] suffered from segmentation failures, while the proposed model was based on the area and achieved better performance in low-contrast cases. In addition, the displayed segmentation results in [57] were not enough; the reason why the segmentation results were affected is not clear. As shown in Table 9, in the DRISHTI-GS dataset, the proposed algorithm also achieved the best average FSE, with 5 points being the most and 0 points being the least. As shown in Table 10, in the TMUEH dataset with the best average FSE, the most cases with 3 or more points and the least cases with 0 points were obtained.

As shown in Table 4, the segmentation results in the DRISHTI-GS dataset were significantly more accurate than those in the TMUEH dataset. There are two main reasons for this:1.Different light sources were used when posterior and wide-angle fundus images are taken. Posterior fundus images use white light sources, while wide-angle fundus images utilize red and green light sources. The proposed method utilizes the value channel in the HSV color space to segment the OD. The value channel is the maximal value of the red, green, and blue of these three channels. However, there is no blue channel information in wide-angle fundus images, which may reduce the segmentation accuracy.2.The resolution of the ROI on posterior fundus images is about 600×600, while the resolution of the ROI on wide-angle fundus images is about 200×200. The low resolution of wide-angle fundus images may also be one of the reasons for the low segmentation accuracy.

However, as shown in Figure 25, there are still some problems that cannot be solved with multiple preprocessing HLSM:1.As shown in Figure 25a, the existence of too-strong blood vessels causes oversegmentation. Because the blood vessels are too thick or multiple blood vessels are entangling, there are still dark shadows after noise removal. The pixel values covered by blood vessels were lower than those in other areas, resulting in undersegmentation.2.As shown in Figure 25b, if there is a large area of bright noise around the OD, the OD is also undersegmented. This situation is predictable, since the proposed method is an area-based level set model, and the initial value is based on thresholding.3.As shown in Figure 25c, the brightness of the ring area (the area between the OD and OC boundaries) was too low, which caused a large error in the initial-value detection, resulting in oversegmentation.

As shown in Figure 26, some segmentation results of representative cases were compared: from a threshold-based method [18], an active contour-based method [31], and the proposed approach. The proposed multiple preprocessing HLSM was proposed to mainly solve the problems of OD segmentation being affected by PPA and bright noise. The other approaches could not achieve ideal results. In addition, problems such as the low brightness of the ring area and large bright noise also cannot be solved by other methods. Therefore, the segmentation results demonstrate that the proposed approach improved the accuracy of the OD segmentation results.

**Figure 25 sensors-22-06899-f025:**
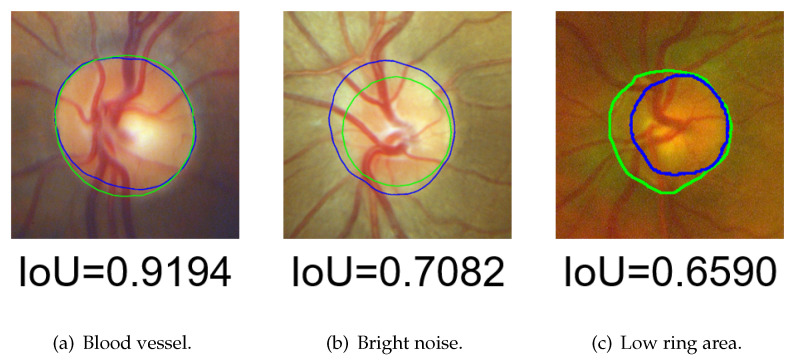
Unsolved problems in OD segmentation (green contour: GT, blue contour: segmentation results).

**Figure 26 sensors-22-06899-f026:**
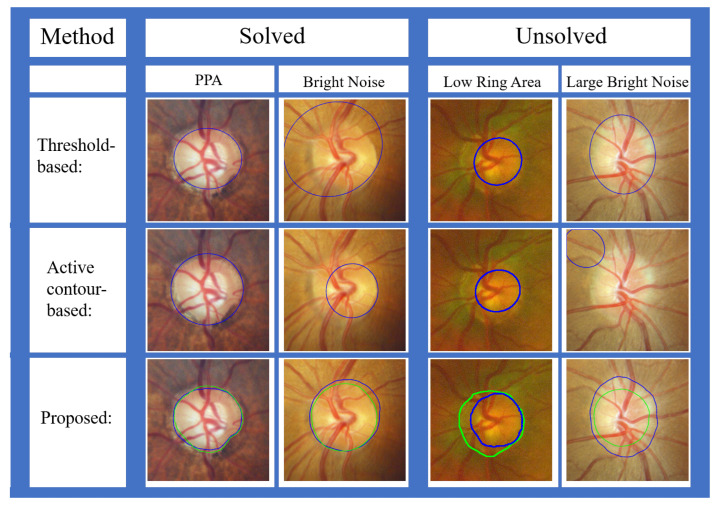
Comparison of some representative cases (green contour: GT, blue contour: segmentation results).

## 5. Conclusions

In this paper, the effectiveness of the proposed multiple preprocessing HLSM was verified in both posterior and wide-angle fundus images. Furthermore, the segmentation results were evaluated objectively with IoU and subjectively with FSE. The proposed approach achieved the following results: average IoU of 0.9275 and average FSE of 4.6426 in the DRISHTI-GS posterior fundus image dataset, and average IoU of 0.8179 and average FSE of 3.5946 in the TMUEH wide-angle fundus image dataset. The proposed multiple preprocessing HLSM solves the effect of PPA regions and bright noise in OD segmentation. This is the first time to segment OD from wide-angle fundus images. The FSE is proposed to partially evaluate OD segmentation results and prove that OD segmentation results are clinically meaningful.

In HLSM, the optimization of parameters is time-consuming, and even though the quality of each image is different, their parameters are same. Therefore, in the future, an automatic parameter optimization method could be proposed, and each image could automatically generate the most suitable parameters according to their quality.

## Figures and Tables

**Figure 1 sensors-22-06899-f001:**
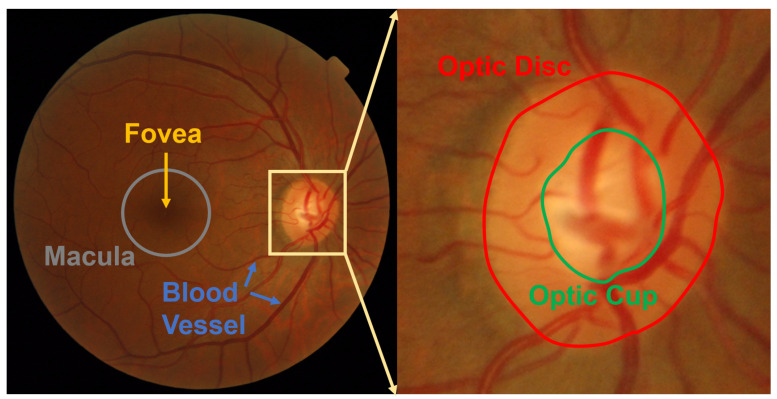
Example of posterior fundus image.

**Figure 2 sensors-22-06899-f002:**
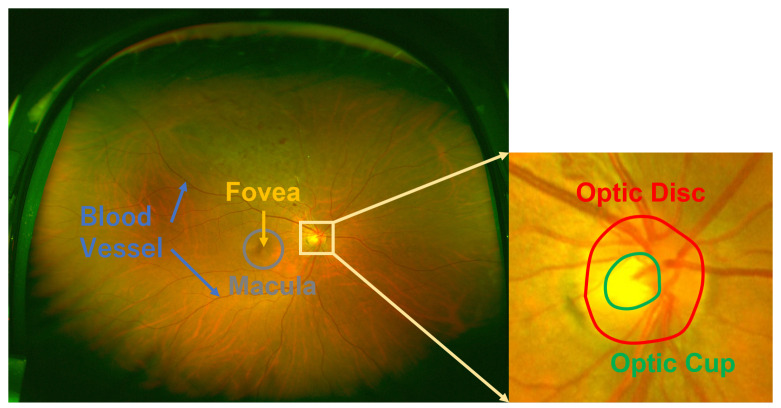
Example of wide-angle fundus image.

**Figure 3 sensors-22-06899-f003:**
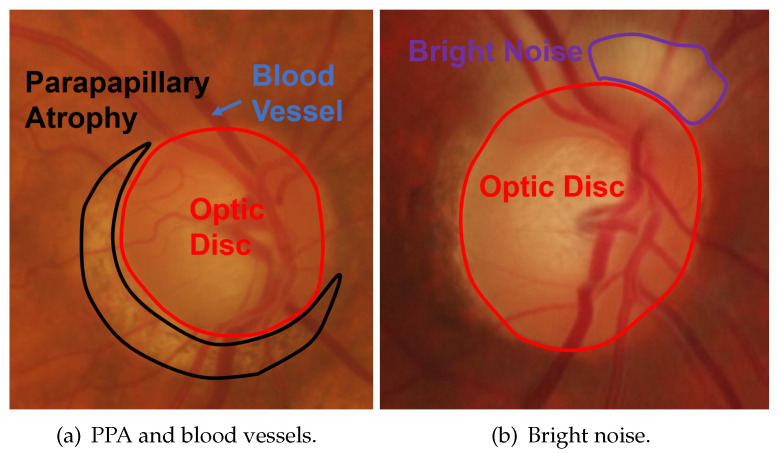
Problems in OD segmentation.

**Figure 4 sensors-22-06899-f004:**
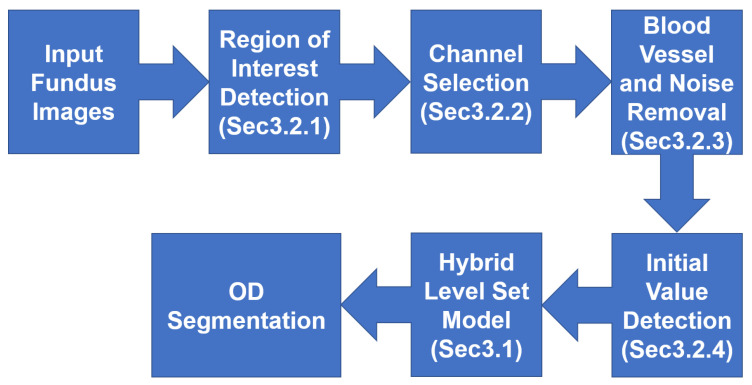
The flowchart of the proposal.

**Figure 5 sensors-22-06899-f005:**
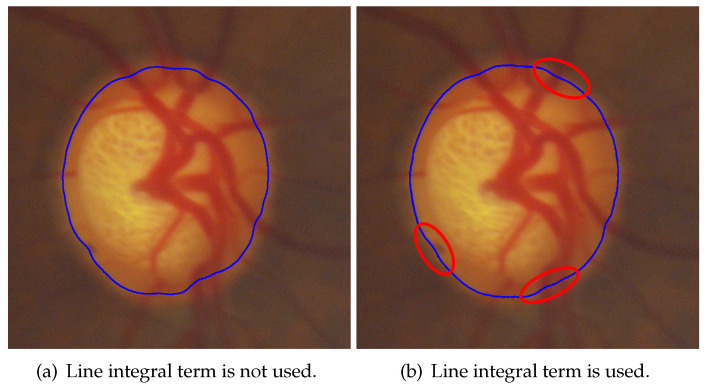
Comparison of using and not using a line integral term; (**b**) the contour was significantly smoother than that in (**a**). (blue contour: results of segmentation, red area: obviously smoother part).

**Figure 6 sensors-22-06899-f006:**
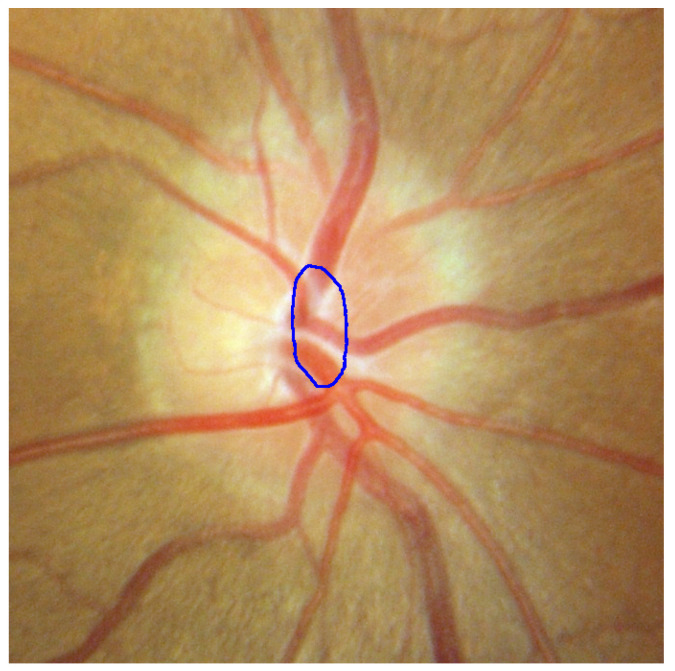
Segmentation result with the gradient-based level set model.

**Figure 7 sensors-22-06899-f007:**
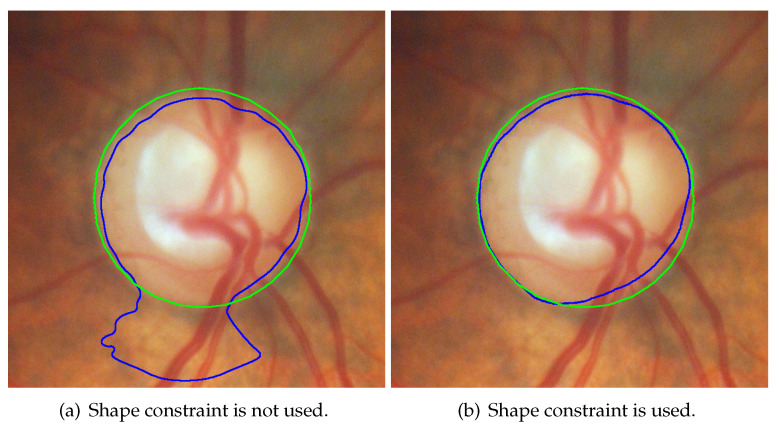
Comparison of shape-based term is used and not used. (blue contour: the segmentation result, green contour: the ground-truth of OD).

**Figure 12 sensors-22-06899-f012:**
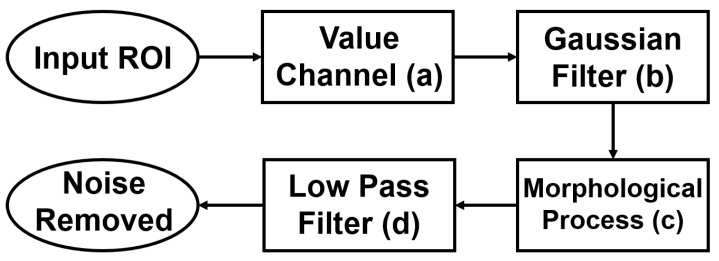
Flowchart of blood-vessel and noise removal ((**a**–**d**) correspond to Figure 13).

**Figure 13 sensors-22-06899-f013:**
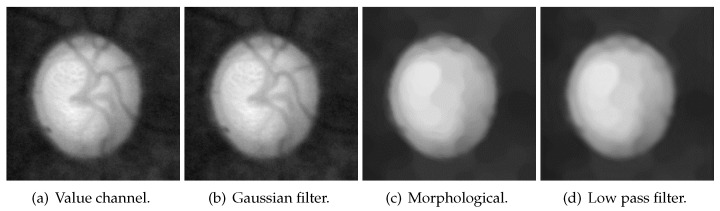
The result of each process in blood-vessel and noise removal.

**Figure 14 sensors-22-06899-f014:**
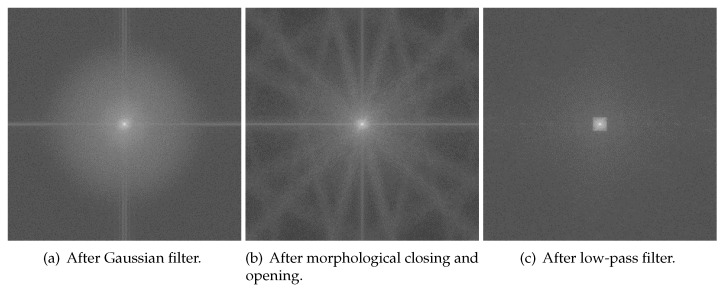
Frequency domain image after each process.

**Figure 15 sensors-22-06899-f015:**
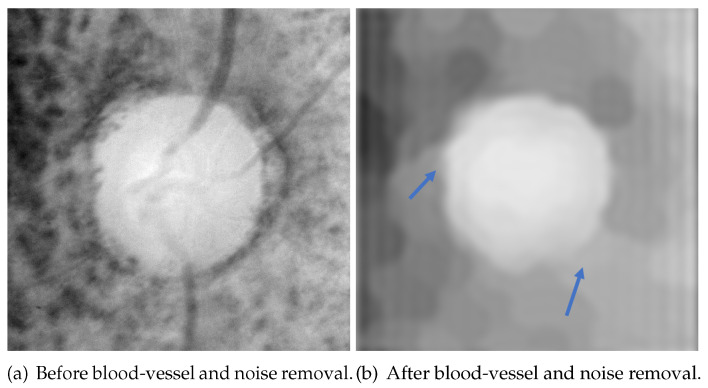
After blood-vessel and noise removal, bright noise (parts indicated by blue arrows) is generated, rendering the OD boundaries not obvious.

**Figure 16 sensors-22-06899-f016:**
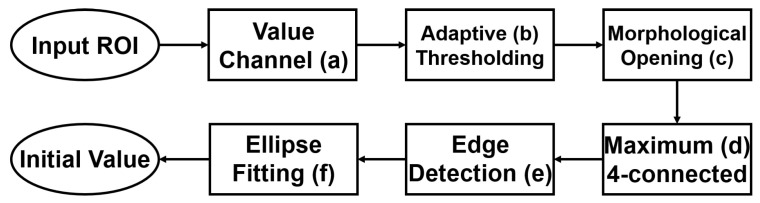
The flowchart of initial-value detection ((**a**–**f**) correspond to the images in Figure 17).

**Figure 17 sensors-22-06899-f017:**
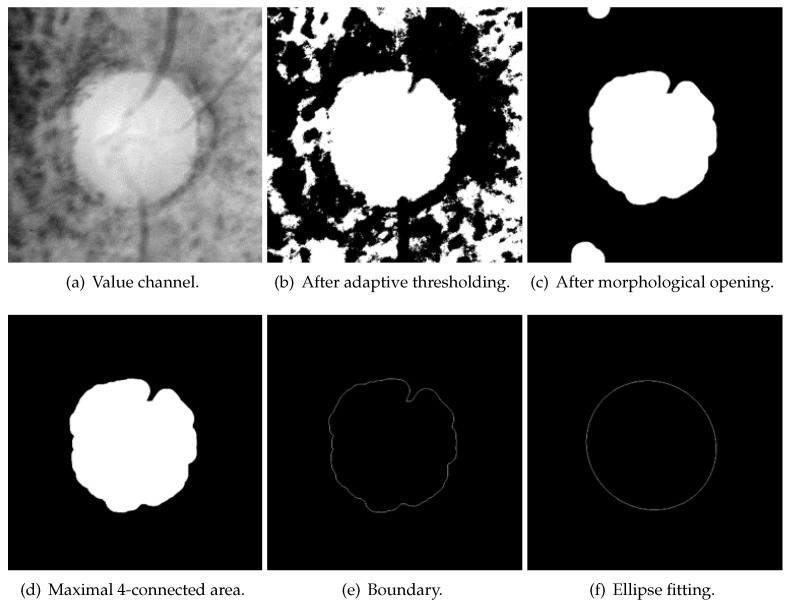
The results of initial-value detection.

**Figure 18 sensors-22-06899-f018:**
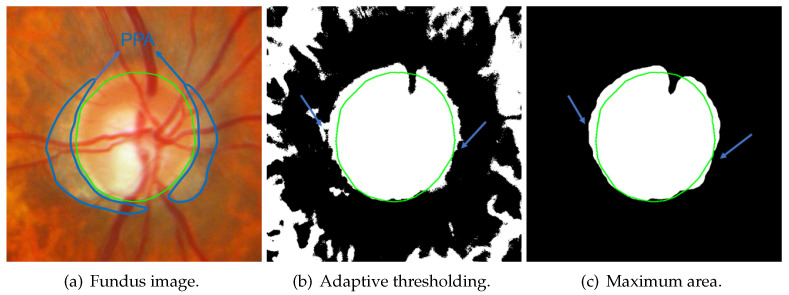
Adaptive thresholding can ignore some PPAs (green contour: ground truth, blue curves and arrows: PPA area. In (**b**,**c**), some PPA regions are ignored).

**Figure 19 sensors-22-06899-f019:**
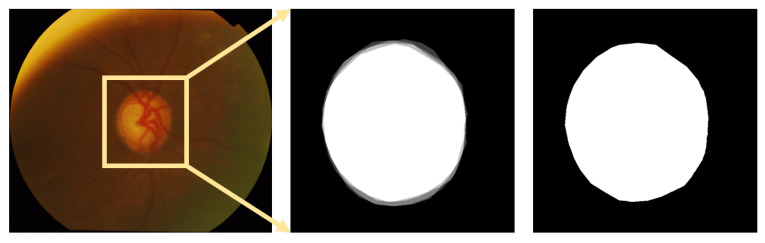
An example of the DRSHTI-GS dataset. (left to right) Original fundus image, GT soft map, and the GT used in this paper (In a soft map, the pixel value of each annotation is 0.25. The part with a pixel value greater than or equal to 0.75 was used as the GT).

**Figure 20 sensors-22-06899-f020:**
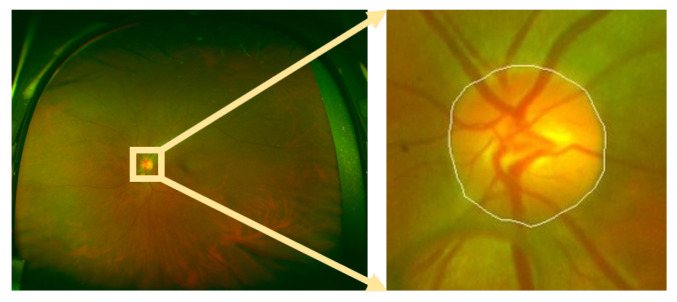
An example of the TMUEH dataset: (left to right) original fundus image and the GT of the OD boundary.

**Figure 21 sensors-22-06899-f021:**
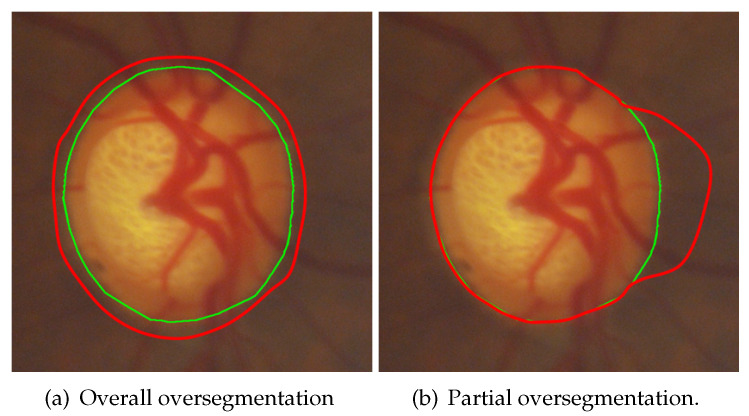
Limitations of IoU evaluation methods, (green contour: GT, red contour: hypothetical segmentation results).

**Figure 22 sensors-22-06899-f022:**
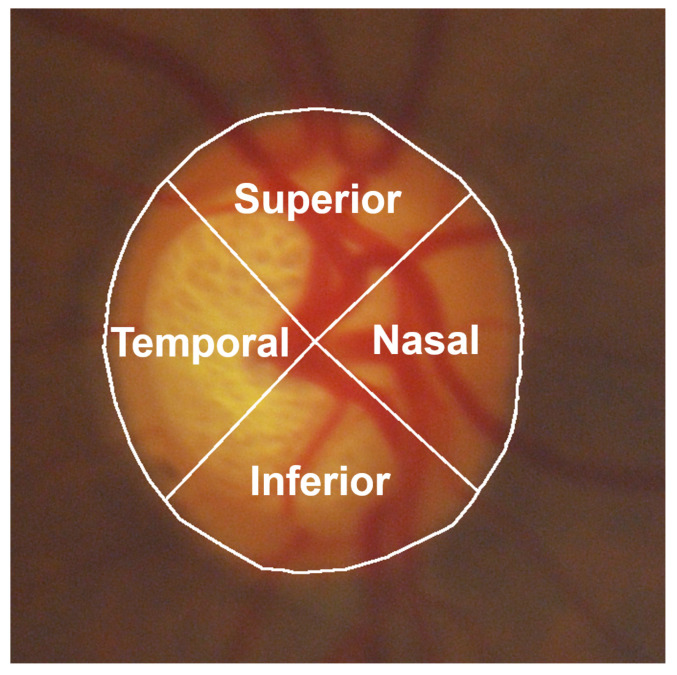
The four parts of OD.

**Figure 23 sensors-22-06899-f023:**
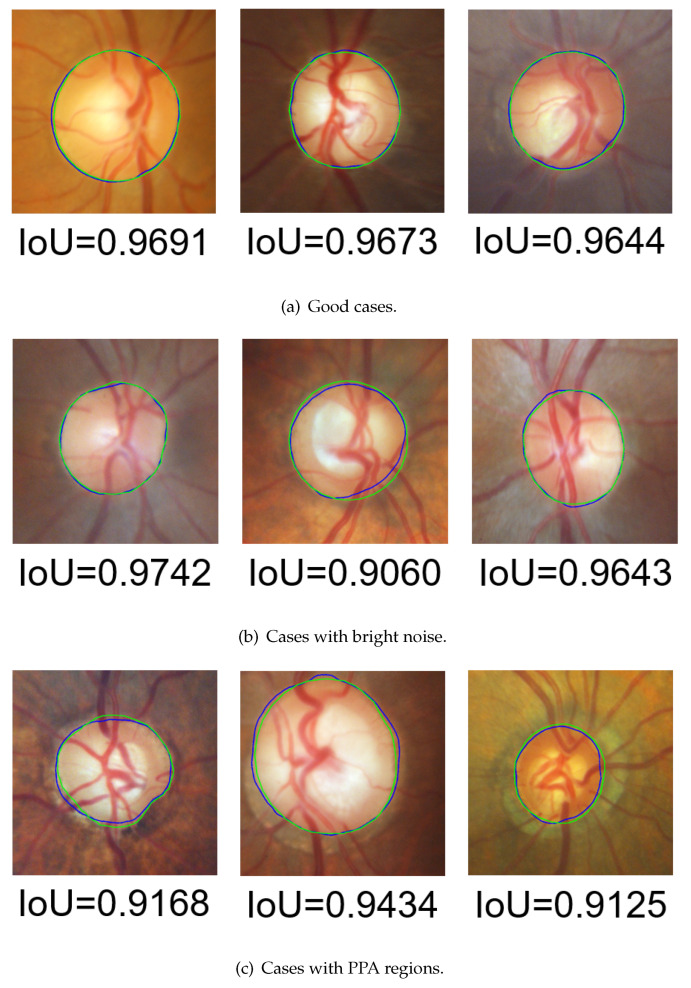
Segmentation result examples in the DRISHTI-GS dataset (green contour: GT, blue contour: segmentation results).

**Figure 24 sensors-22-06899-f024:**
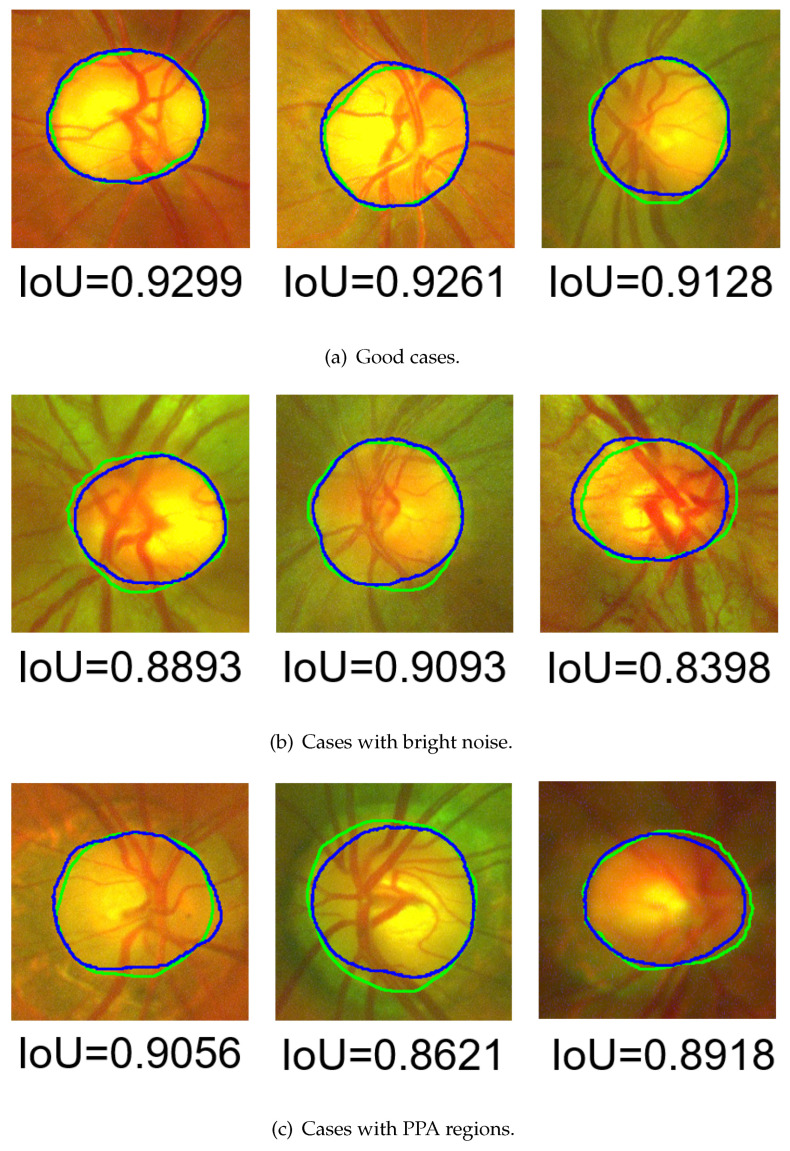
Segmentation result examples in TMUEH dataset (green contour: GT, blue contour: segmentation results).

**Table 1 sensors-22-06899-t001:** Average and variance of CNR values of each channel.

Channel	Average	Variance
Red	4.9824	5.4550
Blue	4.4111	11.6290
Green	4.6481	5.9764
Hue	1.8755	10.1563
Saturation	2.1173	3.0322
**Value**	**5.0408**	5.4443
Lightness	4.6617	5.1113
A (green/magenta)	2.0801	2.5126
B (blue/yellow)	2.6658	4.4554

**Table 2 sensors-22-06899-t002:** The evaluation criteria of FSE.

Score	Criteria
0	There are obvious errors in the boundaries of the four sides.
1	There are errors in the boundaries of four sides, but better than 0.
2	Only one side is accurate enough ^*^
3	Two sides are accurate enough.
4	Three sides are accurate enough.
5	All sides are accurate enough.

* Accurate enough: according to the subjective evaluation of clinician, it would not affect the subsequent diagnosis.

**Table 3 sensors-22-06899-t003:** Parameters used in multiple preprocessing HLSM.

Parameters	Value
The average radius of OD	In posterior fundus images, it is set as the $\frac{1}{3}$ of the radius of the visible circular area. In wide-angle fundus images, it is set as the $\frac{1}{8}$ of the radius of the visible circular area.
μ	0.1
α	3.0
β	0.2
$\lambda_{\mathrm{in}}$	4.3
$\lambda_{\mathrm{out}}$	2.0
$\lambda_{\mathrm{shape}}$	1.1

**Table 4 sensors-22-06899-t004:** Evaluation results with IoU.

Dataset	DRISHTI-GS	TMUEH
Maximal IoU	0.9767	0.9300
Minimal IoU	0.5933	0.5205
Average IoU	0.9275	0.8179
Variance in IoU	0.0025	0.0104
0.9≤ IoU	88/101 cases	8/37 cases
0.8≤ IoU <0.9	11/101 cases	20/37 cases
IoU <0.8	2/101 cases	9/37 cases

**Table 5 sensors-22-06899-t005:** Evaluation results with FSE.

Dataset	DRISHTI-GS	TMUEH
Average FSE	4.6436	3.5946
FSE = 5	83/101 cases	12/37 cases
FSE = 4	8/101 cases	10/37 cases
FSE = 3	5/101 cases	7 /37 cases
FSE = 2	2/101 cases	5/37 cases
FSE = 1	3/101 cases	2/37 cases
FSE = 0	0/101 cases	1 case/37 cases

**Table 6 sensors-22-06899-t006:** Comparison of results in the DRISHTI-GS dataset with those in other papers with average IoU.

Approaches	Average IoU
U-Net [57]	0.8900
BEAL-Deeplabv4+ [60]	0.8620
LARKIFCM [62]	0.9100
U-Net [58]	0.9187
EE-CNN [61]	0.9140
U-Net [59]	0.9062
Proposed	**0.9275**

**Table 7 sensors-22-06899-t007:** Comparison of results in the DRISHTI-GS dataset with reproduced algorithms with IoU.

Dataset	Active Contour-Based [31]	Threshold-Based [18]	Proposed
Maximal IoU	0.9695	0.9720	**0.9767**
Minimal IoU	0	0	**0.5933**
Average IoU	0.8757	0.8760	**0.9275**
Variance in IoU	0.0149	0.0255	**0.0025**
0.9≤ IoU	58/101 cases	74/101 cases	**88/101 cases**
0.8≤ IoU<0.9	29/101 cases	17/101 cases	11/101 cases
IoU<0.8	14/101 cases	10/101 cases	**2/101 cases**

**Table 8 sensors-22-06899-t008:** Comparison of results in the TMUEH dataset with reproduced algorithms with IoU.

Dataset	Active Contour-Based [31]	Threshold-Based [18]	Proposed
Maximal IoU	0.9425	0.9419	0.9300
Minimal IoU	0.1711	0.4671	**0.5205**
Average IoU	0.7321	0.7614	**0.8179**
Variance in IoU	0.0330	0.0141	**0.0104**
0.9≤ IoU	3/37 cases	4 /37 cases	**8/37 cases**
0.8≤ IoU<0.9	16/37 cases	15/37 cases	20/37 cases
IoU<0.8	18/37 cases	18/37 cases	**9/37 cases**

**Table 9 sensors-22-06899-t009:** Comparison of the results in the DRISHTI-GS dataset with reproduced algorithms with FSE.

Method	Active Contour-Based [31]	Threshold-Based [18]	Proposed
Average FSE	4.3069	4.3762	**4.6436**
FSE = 5	60/101 cases	74/101 cases	**83/101 cases**
FSE = 4	19/101 cases	14 /101 cases	8/101 cases
FSE = 3	17/101 cases	3/101 cases	5/101 cases
FSE = 2	4/101 cases	3/101 cases	2/101 cases
FSE = 1	0/101 cases	1 case/101 cases	3/101 cases
FSE = 0	1 case/101 cases	6/101 cases	**0/101 cases**

**Table 10 sensors-22-06899-t010:** Comparison of the results in the TMUEH dataset with the reproduced algorithms by FSE.

Method	Active Contour-Based [31]	Threshold-Based [18]	Proposed
Average FSE	3.4595	3.5135	**3.5946**
FSE = 5	12/37 cases	14/37 cases	12/37 cases
FSE = 4	11/37 cases	6 /37 cases	10/37 cases
FSE = 3	3/37 cases	7/37 cases	7/37 cases
FSE = 2	7/37 cases	7/37 cases	5/37 cases
FSE = 1	1 case/37 cases	1 case/37 cases	2/37 cases
FSE = 0	3/37 cases	2/37 cases	**1 case /37 cases**

## Data Availability

The DRISHTI-GS database in this study is openly available at https://cvit.iiit.ac.in/projects/mip/drishti-gs/mip-dataset2/enter.php accessed on 12 October 2020.

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
