# Peer review of "Multiple Preprocessing Hybrid Level Set Model for Optic Disc Segmentation in Fundus Images"

_sensors, 2022, doi:10.3390/s22186899_

Round 1
Reviewer 1 Report
The authors proposed a multiple pre-processing hybrid level set model based on area and shape for OD segmentation. The work is interesting and needs revision to improve it further. I have the below comments for further improving the quality:
1.Some typos need to be fixed carefully before next revision.
2.Suggest to provide more details of hybrid level set model. Why does the author consider this model instead of others?
3. Figures 5/7/13/14/15/20/23/24/25 would be good if improved further on the visualization, including providing smaller subtitle sizes, aligning the image fonts with the main text, etc.
4.What does Figure 10 “(a)-(i) are same mean with Figure 6” mean? There are 9 images in Figure 10 while only 1 image in Figure 6. Please check whether it is correct.
5.The items of the first and the second lines in Talbe 1 overlap.
Author Response
Dear reviewer 1:
We gratefully appreciate for your valuable suggestions and comments which are really helpful for us to improve the manuscript.
Best regards
Du

Reviewer 2 Report
The authors proposed a novel multiple pre-processing hybrid level set model for OD segmentation. Both IoU and newly proposed FSE were used as model segmentation performance evaluation metrics on a narrow angle and a wide angle fundus image datasets. The authors reported an improved segmentation performance based on IoU measurements by comparing with other studies. The topic is interesting and clinical relevant. However, the text should undergo further proofreading for errors and grammar. The English needs to be improved.
These are some detailed comments:
Abstract:
1. “However, the problems such as vascular occlusion, parapapillary atrophy (PPA), low contrast and so on, accurate OD segmentation is still a challenging task.” Should be “However, because of the problems such as vascular occlusion, parapapillary atrophy (PPA), and low contrast, accurate OD segmentation is still a challenging task.”
2. “in the public dataset gathered narrow-angle fundus images” should be “on a public dataset with narrow-angle fundus images”
Introduction:
1. “is a challenging task as the following reasons” should be “is a challenging task for the following reasons”
2. “uses mathematical morphology to remove blood vessels” is not a complete sentence. It should be “blood vessels can be removed by using mathematical morphology”
3. “HLSM” should be “hybrid level set model (HSLM)”
Related works:
1. “apply the active shape model to segment the OD, but it is easily effected by the PPA and bright noise.” should be rewritten as “Segmenting the OD with the active shape model can be easily effected by the PPA and bright noise.”
Multiple pre-processing hybrid level set
1. “Blue narrows” in Figure 15 caption should be “Blue arrows”
Experiments:
1. Should these images in Figure 21 be over segmentation examples?
Results:
1. Should “Various of IoU” in Table 4 be “variance of IoU”?
2. Did the authors implement the two models, the active contour-based method [31], and the threshold-based method [18], previous reported by other groups, or requested the models from the authors?
3. The proposed model performance is slightly better than other reported models based on the values in Table 6. The improvement may not be statistically significant since there is no statistical analysis performed. The authors need tune down their claims.
4. Do authors have IoU results using the active contour-based method [31], and the threshold-based method [18] from other studies for table 6?
5. Report IoU results on DRISHTI-GS dataset and TMUEH dataset using the active contour-based method [31], and the threshold-based method, similar to Table 7 and Table 8
Author Response
Dear reviwer2
We gratefully appreciate for your careful check and constructive suggestion.
Best regards
Du

Reviewer 3 Report
In this paper, authors a hybrid level set model for optic disc segmentation in fundus images. The proposed method is composed of some parts: ROI detection, channel selection, blood vessel and noise removal, initial value detection, and hybrid level model. To evaluate the performance of the proposed method, two image datasets are used. Experimental results show that the proposed method can provide a good performance.
There are come comments below:
1. Paper presentation could be improved. For example, some symbols are not described clearly.
2. As shown in Fig. 8, authors select 20 circles. It would be better to explain why 20 circles are selected for ROI detection.
3. As shown In Fig. 12, there are two filters: Gaussian filter and low-pass filter. It would be better to explain how to design the two filters.
4. It would be better to explain how to connect the two parts: initial value detection and hybrid level model.
5. As shown in Table 6, authors make a comparison with some existing methods. It would be better to explain why the proposed method outperforms the existing methods [58][61].
Author Response
Dear reviewer 3
We gratefully appreciate for your valuable suggestions and comments.
Best regards
Du
